# Could the Improvement of Supraspinatus Muscle Activity Speed up Shoulder Pain Rehabilitation Outcomes in Wheelchair Basketball Players?

**DOI:** 10.3390/ijerph20010255

**Published:** 2022-12-24

**Authors:** Giacomo Farì, Marisa Megna, Maurizio Ranieri, Francesco Agostini, Vincenzo Ricci, Francesco Paolo Bianchi, Ludovica Rizzo, Eleonora Farì, Lucrezia Tognolo, Valerio Bonavolontà, Pietro Fiore, Victor Machado Reis

**Affiliations:** 1Department of Translational Biomedicine and Neuroscience (DiBraiN), Aldo Moro University, 70121 Bari, Italy; 2Department of Biological and Environmental Science and Technologies (Di.S.Te.B.A.), University of Salento, 73100 Lecce, Italy; 3Department of Anatomical and Histological Sciences, Legal Medicine and Orthopedics, Sapienza University, 00185 Rome, Italy; 4Physical and Rehabilitation Medicine Unit, Luigi Sacco University Hospital, ASST Fatebenefratelli-Sacco, 20157 Milan, Italy; 5Department of Interdisciplinary Medicine, Aldo Moro University of Bari, 70121 Bari, Italy; 6Complex Unit of Territorial Psychology, Department of Mental Health and Pathological Addictions, 40123 Bologna, Italy; 7Physical Medicine and Rehabilitation Unit, Department of Neuroscience, Padua University, 35128 Padua, Italy; 8Department of Biotechnological and Applied Clinical Sciences, University of L’Aquila Vetoio, 67100 L’Aquila, Italy; 9Istituti Clinici Scientifici Maugeri, IRCCS Institute of Bari, 70121 Bari, Italy; 10Research Centre in Sport Sciences, Health Sciences and Human Development, 5001-801 Vila Real, Portugal

**Keywords:** shoulder pain, wheelchair basketball, Paralympics, sport, rehabilitation, disability, rotator cuff

## Abstract

Shoulder pain (SP) is a common clinical complaint among wheelchair basketball (WB) players, since their shoulders are exposed to intense overload and overhead movements. The supraspinatus tendon is the most exposed to WB-related injuries and it is primarily responsible for SP in WB athletes. In these cases, SP rehabilitation remains the main treatment, but there is still a lack of specific protocols which should be customized to WB players’ peculiarities and to the supraspinatus muscle activity monitor, and the improvement of rehabilitation outcomes is slow. Thus, the aim of this study was to verify if the improvement of supraspinatus muscle activity, monitored in real time with surface electromyography (sEMG) during the execution of therapeutic exercises, could speed up SP rehabilitation outcomes in WB players. Thirty-three athletes were enrolled. They were divided into two groups. Both groups underwent the same shoulder rehabilitation program, but only the Exercise Plus sEMG Biofeedback Group executed therapeutic exercises while the activity of the supraspinatus muscles was monitored using sEMG. Participants were evaluated at enrollment (T0), at the end of 4 weeks of the rehabilitation program (T1), and 8 weeks after T1 (T2), using the following outcome measures: supraspinatus muscle activity as root mean square (RMS), Wheelchair User’s Shoulder Pain Index (WUSPI), shoulder abduction, and external rotation range of motion (ROM). The Exercise Plus sEMG Biofeedback Group improved more and faster for all the outcomes compared to the Exercise Group. The monitoring and improvement of supraspinatus muscle activity seems to be an effective way to speed up SP rehabilitation outcomes in WB players, since it makes the performance of therapeutic exercise more precise and finalized, obtaining better and faster results in terms of recovery of shoulder function.

## 1. Introduction

Wheelchair basketball (WB) is a Paralympic sport which grants many benefits for people with disabilities who practice it [1]. WB is a contact sport and requires intense sport gestures and athletic performance, in which frequent shoulder movements are necessary (such as quick direction changes and sprints, ball throwing and passing); thus, it is unavoidable that this sport exposes players to injuries [2], particularly in the shoulders, so much so that shoulder pain (SP) is extremely common among these athletes [3]. As a consequence, an athlete may need to stop the sport activity, remaining inactive for a period ranging from days to months and suffering both physical and psychological negative repercussions [4]. SP also results in a limitation of the activities of daily living in wheelchair users, and the need to regularly use the wheelchair prevents the possibility of rest aimed at a full functional recovery, triggering a vicious cycle.

It follows that SP in disabled athletes is a problem of constant relevance in the field of sports medicine and rehabilitation. A recent scoping review states that SP prevalence in WB players can reach the threshold of 75% [5]. Among the SP causes in WB players, rotator cuff tendinopathies seem to be the most frequent trigger [6]. In particular, since WB is an overhead sport, the supraspinatus tendon is the most exposed to lesions and thus is primarily responsible for the appearance of SP [7], also contributing to scapular dyskinesias, which results in further limitations of daily wheelchair use [8]. SP rehabilitation in WB athletes remains the main therapeutic approach. Nevertheless, SP rehabilitation protocols for WB players are still not specific and customized to the needs of these athletes and to supraspinatus muscle activity monitoring, and are therefore characterized by slow outcome improvement [9].

Surface electromyography (sEMG) allows measuring the muscles’ electrical activity and identifying any alterations, correcting them in real time [10]; thus, it represents a promising instrument to personalize kinesitherapy programs. We hypothesized that EMG monitoring of rehabilitation can more rapidly improve joint function and reduce SP.

Thus, the aim of this study was to verify if the improvement of supraspinatus muscle activity, monitored in real time with sEMG during the execution of therapeutic exercises, could speed up SP rehabilitation outcomes in WB players and guarantee them a quicker return to the field, avoiding a prolonged suspension of sporting activity.

## 2. Materials and Methods

Between May 2021 and September 2022, a prospective clinical study was carried out at the Movement Analysis Service of the Department of Biological and Environmental Science and Technology, University of Salento, Lecce, Italy.

### 2.1. Participants

The following enrollment criteria were established. Inclusion criteria: athletes aged >18 years; members of a professional WB sports association; sporting practice for at least 2 years; prevalent wheelchair use in the activities of daily living; SP for at least 1 month; medical and ultrasound diagnosis of rotator cuff tendinopathy. Exclusion criteria: SP conservative or surgical therapies in the previous month; presence of shoulder fractures and arthropathies; presence of complete rotator cuff tendon tears; clinical or instrumental evidence of rheumatological or neurological diseases affecting the upper limbs. WB athletes participating in the second Italian divisions of FIPIC (Wheelchair Basketball Italian Federation) were eligible for recruitment, provided that they met the above-mentioned criteria.

Thirty-three (33) athletes met these criteria and were recruited for the study. The sample size was a convenience one, but it was in line with previous studies concerning biofeedback and wheelchair athletes’ rehabilitation [11,12]. G*Power post hoc calculations for the performed ANOVA indicated a statistical power of 96%, provided a minimum effect size of 0.3 (as given by eta squared).

### 2.2. Procedures

At the enrollment (T0), after a medical evaluation, each participant underwent the following evaluations:-WUSPI (Wheelchair User’s Shoulder Pain Index): It is a scale which measures shoulder pain associated with the functional activities of wheelchair users. This 15-item index investigates shoulder pain during transfers, self-care, wheelchair mobility, and general activities. The score can range from 0 to 150 [13].-Supraspinatus muscle activity of both shoulders measured as root mean square (RMS): This evaluation was made using the mDurance sEMG and was expressed in microvolts; after performing a skin disinfection with isopropyl alcohol and placing the patient in a shoulder resting position, two pre-gelled electrodes were applied on two points of the supraspinatus muscle belly and one third on the acromion, for both sides, according to the pre-set device instructions.-Range of motion (ROM) in abduction and external rotation: It is the evaluation measured in degrees (°) of the shoulder range of movement in the direction most influenced by the activity of the supraspinatus muscle. This evaluation was made using the inertial sensors included in the mDurance device.

Then, all the recruited WB players were randomly divided into two groups: the Exercise Group and the Exercise Plus sEMG Biofeedback Group. Both groups underwent a shoulder rehabilitation protocol under the guidance of a therapist according to a four-week exercise protocol (two exercise sessions per week, one hour per session). Exercises’ volume and intensity were adapted for each individual within a pain-free range, but each participant was asked to perform at least two series of at least eight repetitions for each exercise. Exercises focused on stretching and strengthening of shoulder rotators, adductors, abductors, and extensors, with particular attention to supraspinatus. Resistance exercises involved the initial use of rubber bands and then the use of increasing weights, from 1 to 3 kilograms, over the course of the 4-week program. Particularly, exercises consisted of shoulder rotations in the horizontal plane, shoulder abduction and adduction with and without weights, and shoulder and elbow flexion and extension with and without weights. At the end of each session, stretching exercises were carried out for all the muscles of the shoulder girdle.

The difference between the groups was that subjects in the exercise plus sEMG Biofeedback group executed all the exercises under the control of the mDurance^®^ system [14], which made it possible for the therapist and the athlete to activate and monitor the supraspinatus muscles in real time, using sEMG biofeedback to improve shoulder muscle balance and muscle relaxation. The mDurance^®^ system consists of a two-channel bipolar sensor for recording the superficial muscle activity. This electrical signal is brought from the pre-gelled electrodes, which are applied on the skin in correspondence with the investigated muscle, to a tablet, where a dedicated mobile application stores, analyzes, and transforms the signal into live images visible on the tablet screen. So, during the exercise execution, both the patient and the therapist can look at the graphs of muscle activation and joint mobility, working on a model of self-correction in biofeedback and also obtaining a final report about the performance. 

Each enrolled subject was newly evaluated at T1, at the end of the rehabilitation protocol, 4 weeks after T0, and at T2, 8 weeks after T1. To guarantee a pharmacological pain control, in the twelve weeks after T0, athletes were allowed to take paracetamol as needed (maximum 3 g/day) and they were asked to report the frequency and dosage in an intake diary.

Each participant was recruited with informed consent to participate in the study. All the procedures were carried out in accordance with the principles of the Declaration of Helsinki. Ethical approval was granted by the Institutional Review Board of the University of Salento (no. 3/28 April 2021).

### 2.3. Statistical Analysis

Compiled forms were entered into a database created with an Excel spreadsheet, and data analysis was performed using Stata MP17 software. Continuous variables were described as mean ± standard deviation (SD) and range, and categorical variables as proportions. The skewness and kurtosis test was used to evaluate the normality of continuous variables; all the continuous variables were normally distributed. Student’s t test for independent data was used to compare continuous variables between groups. The ANOVA for repeated measures test was used to compare continuous variables between groups and detection time. A post hoc analysis was performed using the test of simple effects to estimate the variation in each outcome confronting each detection time per group. The chi-square test was used to compare the proportions between groups. For all tests, a two-sided *p*-value < 0.05 was considered statistically significant.

## 3. Results

The study sample included 33 male subjects, of which 16 (48.5%) belonged to the Exercise Group and 17 (51.5%) to the Exercise Plus sEMG Biofeedback Group. The characteristics of the sample, by group, are shown in Table 1. The groups were homogeneous according to the considered variables.

The mean ± SD and range of the outcome variables, by group and time of detection, are described in Table 2 and graphically represented in Figure 1. Supraspinatus RMS of the limb affected by SP increased in both groups between the three detection times, but more markedly in the Exercise Plus sEMG Biofeedback Group, while supraspinatus RMS of the limb free from SP slightly increased between times, with no substantial differences between the two groups. WUSPI scores improved in both groups between the three detection times, but more markedly in the Exercise Plus sEMG Biofeedback Group. Abduction and External Rotation ROM scores improved in both groups between the three detection times, but more markedly in the Exercise Plus sEMG Biofeedback Group.

The ANOVA for repeated measures test showed a statistically significant difference for supraspinatus RMS of the limb affected by SP, WUSPI, external rotation, and abduction ROM scores in the comparison between groups (*p* < 0.05), while supraspinatus RMS for the limb free from SP did not show a significant difference in this comparison (*p* = 0.844); there was a statistically significant difference for all the outcome measures in the comparison between times (*p* < 0.0001). The same test showed a statistically significant difference for all the outcome measures in the interaction between time and group (*p* < 0.0001), except the supraspinatus RMS for the limb free from SP (0.686). So, the Exercise Plus sEMG Biofeedback Group showed a better improvement for all the outcomes in the detection times compared to the Exercise Group. All these findings are described in Table 2.

In Table 3, a statistically significant improvement in the supraspinatus RMS for both shoulders emerged for both groups between T0 and T1 (*p* < 0.0001), between T0 and T2 (*p* < 0.0001), and between T1 and T2 (*p* < 0.0001 for Exercise Plus sEMG Biofeedback Group and *p* = 0.004 for Exercise Group), so the improvement for the Exercise Plus sEMG Biofeedback Group was higher. Neither group recorded significant differences between T1 and T2 for supraspinatus RMS for the limb free from SP (*p* = 0.066 for Exercise Plus sEMG Biofeedback Group and *p* = 0.061 for Exercise Group). A statistically significant improvement in the WUSPI scores emerged for both groups between T0 and T1, between T0 and T2, and between T1 and T2 (*p* < 0.05). The abduction and external rotation ROM scores improved in both groups between all the detection times, but these differences are higher for the Exercise Plus sEMG Biofeedback Group.

From the analysis of analgesic intake diaries in the twelve weeks following enrollment, only a random intake emerged, which settled on an average of 1.5 g/week per group, with a sporadic and non-significant distribution among the participants.

## 4. Discussion

Sporting activities are a great health occasion for people with disabilities [15,16], but sport-related injuries are a frequent risk both for able-bodied and disabled athletes [17,18,19,20,21,22]. In particular, WB players are maximally exposed to SP [23]. Thus, this clinical complaint requires the identification of new prevention and therapy strategies. Rehabilitation is a traditional effective approach to relieve SP for wheelchair users and wheelchair athletes [24,25], but there is still the necessity to better standardize rehabilitative protocols in order to speed up and better monitor the improvement of outcomes, especially for WB players, whose peculiarities in terms of sport gestures require more specific, targeted, and faster therapeutic solutions than those already existing. Since rotator cuff dysfunctions, particularly supraspinatus muscle ones, are the most frequent causes of WB players’ SP, the aim of this study was to elucidate if real-time monitoring and specific improvement of supraspinatus muscle activity could speed up and better target SP rehabilitation in this category of disabled sportsmen.

The results showed that both groups improved, but the Exercise Plus sEMG Biofeedback Group obtained better and faster results for all the outcome measures in all the detection times compared to the Exercise Group. In particular, a higher improvement of supraspinatus RMS for the upper limb affected by SP coincides with a greater and faster joint ROM increase and pain relief. The possibility to monitor supraspinatus muscle activity in real time using sEMG seems to be an effective way to finalize and guide the execution of therapeutic shoulder exercises, focusing on this pivotal muscle as the cause of SP and thus allowing a faster recovery in WB players.

These findings are in line with those reported in other studies regarding therapeutic exercise protocols for SP in wheelchair users. The obtained improvement in WUSPI scores is superimposable to that achieved by Middaugh et al. [26], who carried out EMG biofeedback training, in addition to a traditional exercise protocol, on reducing SP in manual wheelchair users with spinal cord injury. The WUSPI improvement for the experimental group at the 10-week follow-up reached a rate of 64%, similarly to what was achieved in our Exercise Plus sEMG Biofeedback Group, and it was greater than that of the control group, which executed rehabilitation without EMG biofeedback. Furthermore, the increases in abduction and external rotation ROM showed significant improvements, since they were greater than 5° [27], and were preventable as a consequence of a rehabilitation plan aimed at recovering joint function starting from the restoration of the best activity of the supraspinatus muscle, which guarantees and implements these movements when appropriately prompted by targeted exercises [28]. In fact, Wilroy et al. [29] devised a shoulder injury prevention program aimed at improving joint ROM and relieving pain in WB players. This study was carried out on a small sample, but it demonstrated that internal and external rotation significantly increased with rehabilitation programs targeting the rotator cuff muscles.

The improvement of supraspinatus muscle activity is traditionally the keystone of the protocols aimed at full functional recovery from SP deriving from rotator cuff tendinopathies [30], especially for overhead athletes [31]. It is even more important for wheelchair athletes. In this sense, Aytar et al. [8] stated the necessity for disabled athletes of special exercise techniques for shoulder and dyskinesis to be included in training programs to prevent injury. These protocols should aim to train the shoulders symmetrically, thus harmonizing dyskinesias and normalizing the activity of the cuff muscles [32,33]. As is known, SP causes a reduction in daily shoulder movements and therefore in the activation of the relative muscles, while specific training, especially if symmetrical, reactivates the muscles, particularly the supraspinatus, and implements their electrical activity, improving global joint function [34,35,36]. Although the shoulder free from SP was not the specific target of the rehabilitation that we carried out, it is not surprising that this shoulder also improved, progressively matching the values of the contralateral one, while obviously the supraspinatus RMS value increased slightly, since it was already higher at baseline.

The possibility to control and train the supraspinatus muscle in real time using sEMG seems to be a promising way to quickly increase shoulder ROM and to relieve SP, thereby speeding up rehabilitation outcomes and making them easier to measure, thus addressing the problem of objectifying rehabilitative evaluation scales [37,38]. Nevertheless, to our knowledge, sEMG control of shoulder muscles has never been included in rehabilitation protocols dedicated to WB players. However, even when sEMG was implemented in rehabilitation and return-to-field programs for athletes practicing overhead sports, it was mainly used as a mere diagnostic tool for muscle dysfunctions [39,40], while the application we propose is a novelty since it is used as a guide, measurement, and biofeedback correction tool for shoulder therapeutic exercises. We suppose that sEMG traditional applications were limited because it was a complex and expensive instrument, and consequently, it was an exclusive prerogative of expert physicians. However, the latest technological developments in sEMGs make these devices easy and feasible [41,42]. In fact, mDurance has a guide that graphically explains and represents on a tablet screen all the procedures, from applying the sensors to the interpretation of the final reports. In this way, sEMG allows athletes to observe and correct shoulder exercises in biofeedback by specifically implementing the muscle activity of the shoulder girdle muscles, and it also allows them to perform exercises autonomously in home-based protocols which can at a later time be controlled in all aspects by the physiotherapists who treat them. Furthermore, in this way, the physiotherapist can better control muscle activation, increasing it from the point of view of WB-specific gestures or other sport-specific needs. 

This study is not free from limitations. The sample size is a convenience one since WB players are a small cohort themselves. Nevertheless, as we stated above, the sample enabled 96% statistical power and it is in line with other similar studies. As a consequence, this study is not a randomized clinical trial, so further studies are needed, with larger cohorts, to better demonstrate the evidence that the supraspinatus muscle activity improvement could speed up SP rehabilitation outcomes. Finally, the follow-up periods were short, and it is desirable for new studies to investigate the duration of outcome improvement. We also consider the adaption of the volume and intensity of exercise as a study limitation. The modulation of the loads according to the individual abilities of the athletes makes the exercise method less objective, but this need was due to the necessity to remain in a pain-free range of motion and therefore to allow all the exercises to be carried out, as we said before. The individualization of therapeutic exercise represents a further challenge, which will certainly be possible thanks to the use of sEMG. Nevertheless, a strength of this study is that it is the first to place the monitoring and improvement of the supraspinatus activity at the center of the SP rehabilitation project in wheelchair athletes, outlining a new and promising strategy to speed up rehabilitation outcomes.

## 5. Conclusions

The monitoring and improvement of supraspinatus muscle activity seems to be an effective way to speed up SP rehabilitation outcomes in WB players, since it makes the execution of therapeutic exercise more precise and finalized, obtaining better and faster results in terms of recovery of shoulders’ full function and motor harmony. As a consequence, professional athletes can return to the field more quickly, and more generally, people with disabilities who practice WB can just as quickly return to their usual physical activity, avoiding the risks caused by abandoning the sport. Further studies are needed to implement the evidence of the importance of supraspinatus muscle activity monitor and improvement in SP rehabilitation protocols, but this already seems a promising way to build new shoulder sports rehabilitation programs for wheelchair athletes.

## Figures and Tables

**Figure 1 ijerph-20-00255-f001:**
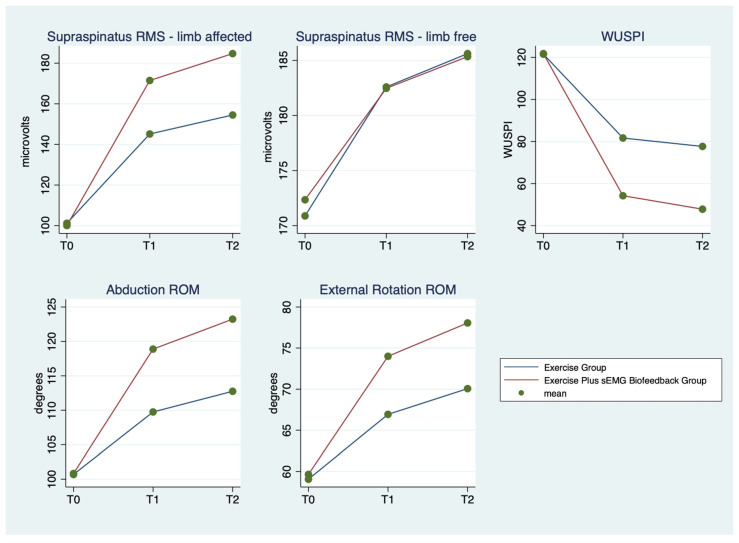
Outcomes’ mean scores by group and detection time.

**Table 1 ijerph-20-00255-t001:** Sample features, by group.

Variable	Exercise Group (n = 16)	Exercise Plus sEMG Biofeedback Group (n = 17)	Total (n = 33)	*p*-Value
Age; mean ± SD (range)	38.7 ± 8.7 (23–55)	37.2 ± 10.0 (24–57)	37.9 ± 9.3 (23–57)	0.648
BMI; mean ± SD (range)	23.7 ± 3.2 (18.8–31.1)	24.4 ± 4.9 (17.0–35.9)	24.0 ± 4.1 (17.0–35.9)	0.623
Right dominant limb; n (%)	13 (81.3)	12 (70.6)	25 (75.8)	0.475
Shoulder pain on the right side; n (%)	14 (87.5)	13 (76.5)	27 (81.8)	0.412

Control = control group; Experimental = experimental group; BMI = Body Mass Index; SD = standard deviation; n = number.

**Table 2 ijerph-20-00255-t002:** Mean ± standard deviation (range) of the outcomes, by group and detection time.

Outcome	Group	T0	T1	T2	Comparison Between Groups	Comparison Between Times	Interaction Between Time and Group
Supraspinatus RMS (microvolts)—limb affected by SP	Exercise Group	101.2 ± 11.9 (81.6–118.9)	145.1 ± 12.9 (123.7–170.0)	154.5 ± 11.8 (132.1–179.3)	<0.0001	<0.0001	<0.0001
	Exercise Plus sEMG Biofeedback Group	100.1 ± 13.1 (78.1–119.1)	171.5 ± 5.1 (160.2–178.9)	184.7 ± 6.7 (170.3–195.7)			
	Total	100.6 ± 12.3 (78.1–119.1)	158.7 ± 16.4 (123.7–178.9)	170.1 ± 18.0 (132.1–195.7)			
Supraspinatus RMS (microvolts)—limb free from SP	Exercise Group	170.9 ± 4.8 (160.3–180.5)	182.6 ± 8.7 (167.9–201.7)	185.6 ± 11.8 (132.1–179.3)	0.844	<0.0001	0.686
	Exercise Plus sEMG Biofeedback Group	172.3 ± 6.0 (158.5–180.0)	182.5 ± 5.2 (172.9–189.1)	185.3 ± 3.8 (178.2–191.0)			
	Total	171.6 ± 5.4 (158.5–180.5)	182.5 ± 7.0 (167.9–201.7)	185.5 ± 6.1 (174.6–200.9)			
WUSPI	Exercise Group	121.6 ± 9.6 (107–137)	81.7 ± 6.3 (71–92)	77.7 ± 6.6 (66–89)	<0.0001	<0.0001	<0.0001
	Exercise Plus sEMG Biofeedback Group	121.8 ± 8.9 (106–135)	54.2 ± 8.5 (41–70)	47.8 ± 8.7 (36–67)			
	Total	121.7 ± 9.1 (106–137)	67.5 ± 15.8 (41–92)	62.3 ± 17.0 (36–89)			
Abduction ROM (degrees)	Exercise Group	100.7 ± 3.4 (97–107)	109.8 ± 2.6 (105–115)	112.8 ± 2.6 (108–118)	<0.0001	<0.0001	<0.0001
	Exercise Plus sEMG Biofeedback Group	100.8 ± 3.8 (94–110)	114.5 ± 5.8 (105–126)	118.2 ± 6.2 (108–128)			
	Total	100.8 ± 4.2 (94–110)	118.9 ± 4.2 (111–126)	123.2 ± 3.7 (116–128)			
External Rotation ROM (degrees)	Exercise Group	59.1 ± 7.9 (47–73)	66.9 ± 7.2 (58–80)	70.1 ± 6.8 (62–82)	0.012	<0.0001	<0.0001
	Exercise Plus sEMG Biofeedback Group	59.6 ± 6.9 (48–72)	74.0 ± 3.2 (70–80)	78.1 ± 2.9 (74–82)			
	Total	59.4 ± 7.3 (47–73)	70.6 ± 6.5 (58–80)	74.2 ± 6.5 (62–82)			

Control = control group; Experimental = experimental group; RMS = root mean square; SP = shoulder pain; WUSPI = Wheelchair User’s Shoulder Pain Index; ROM = range of motion.

**Table 3 ijerph-20-00255-t003:** Time effect at each treatment level.

		Exercise Plus sEMG Biofeedback Group		Exercise Group	
Outcome	Time	Contrast (95%CI)	*p*-Value	Contrast (95%CI)	*p*-Value
Supraspinatus RMS—limb affected by SP	T1 vs. T0	71.4 (65.4–77.5)	<0.0001	43.9 (37.7–50.2)	<0.0001
	T2 vs. T0	84.7 (78.6–90.7)	<0.0001	53.3 (47.1–59.2)	<0.0001
	T2 vs. T1	13.2 (7.2–19.2)	<0.0001	9.4 (3.2–15.6)	0.004
Supraspinatus RMS—limb free from SP	T1 vs. T0	10.1 (7.0–13.2)	<0.0001	11.7 (8.5–14.9)	<0.0001
	T2 vs. T0	13.0 (9.9–16.1)	<0.0001	14.7 (11.6–17.9)	<0.0001
	T2 vs. T1	2.9 (−0.2–6.0)	0.066	3.0 (−0.1–6.2)	0.061
WUSPI	T1 vs. T0	−67.6 (−71.2–−63.9)	<0.0001	−39.9 (−43.6–−36.1)	<0.0001
	T2 vs. T0	−74.0 (−77.6–−70.4)	<0.0001	−43.9 (−47.6–−40.1)	<0.0001
	T2 vs. T1	−6.4 (−10.0–−2.8)	0.001	−4.0 (−7.7–−0.3)	0.037
Abduction ROM	T1 vs. T0	18.1 (16.3–19.8)	<0.0001	9.1 (7.3–10.9)	<0.0001
	T2 vs. T0	22.4 (20.7–24.2)	<0.0001	12.1 (10.3–13.9)	<0.0001
	T2 vs. T1	4.4 (2.6–6.1)	<0.0001	3.0 (1.2–4.8)	0.001
External Rotation ROM	T1 vs. T0	14.3 (12.3–16.4)	<0.0001	7.9 (5.7–10.0)	<0.0001
	T2 vs. T0	18.4 (16.3–20.5)	<0.0001	11.0 (8.8–13.2)	<0.0001
	T2 vs. T1	4.1 (2.0–6.1)	<0.0001	3.1 (1.0–5.3)	0.005

RMS = root mean square; WUSPI = Wheelchair User’s Shoulder Pain Index; ROM = range of motion; CI = confidence interval.

## Data Availability

The datasets used and analyzed during the current study will be made available upon reasonable request to the corresponding author, G.F.

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
