# Peer review of "Could the Improvement of Supraspinatus Muscle Activity Speed up Shoulder Pain Rehabilitation Outcomes in Wheelchair Basketball Players?"

_ijerph, 2022, doi:10.3390/ijerph20010255_

Round 1
Reviewer 1 Report
Dear Authors
The presented article is of great interest to the scientific community. The Methodology presented with rigor, data presented with rigor, and reasoned discussion of the same.
However, what is described in lines 122 to 125 is not clear, so it is important to clarify the criteria for taking paracetamol, how many athletes have taken it, and in what dosages they have taken it. This issue should also be addressed when discussing the data.
Author Response
Dear Sir,
thank You so much for Your comments, it is a great honor for us that You find our study interesting and well presented.
With regard to lines from 122 to 125, we better explained these data according to Your request within all the text.
Best regards
Reviewer 2 Report
This study verified the supraspinatus electromyographic activity, pain index and range of motion after 3 months of a therapy rehabilitation program carried out with and without electromyographic feedback in the shoulder of wheel chair basketball players. The results are promising and shown a better recovery in the athletes undergone therapy rehabilitation with electromyographic feedback. However, the therapy rehabilitation program is poorly and subjectively described. In my opinion, it would not be possible to reproduce the study with the information provided.
Introduction
The introduction justified the relevance for studying shoulder rehabilitation therapies in wheel chair basketball players, but the rationale for why the therapy with electromyographic activity feedback would be more efficient is not provided. Additionally, there is not a hypothesis for the possible outcomes.
Methods
The experimental design seems adequate, but the procedures are not well described and important details are missing. This lack of information precludes a deeper analysis.
There is no other information about the exercise protocol that the participants underwent besides the frequency of twice a week for 3 months. What was the session duration, exercise type, exercise volume, exercise intensity and the adherence of the therapy rehabilitation program throughout the experimental period?
The use of mDurance system was poorly described, there is no information about the technical characteristics, configuration, practical application and handling of the equipment for the specific study. Moreover, the information regarding the signal data processing is also missing.
It is not clear how the feedback provided by the surface EMG was used to implement the supraspinatus activity. In other words, what was the objective difference between the rehabilitation session carried out in the two groups? What was the variable visualized in real time (RMS) during the sessions? What was the parameter used to guide the therapy sessions?
The same procedures were carried out in both shoulders or the procedures where different in the shoulder with and without pain?
The statistical analysis was adequate.
Results
The results are clearly presented.
Discussion
The discussion is a “reprising” of the results. The study would benefit from a possible physiological mechanism to explain the therapeutic exercise effect on the rehabilitation of the outcomes. Moreover, it would be nice to find an explanation for the similar RMS values of the pain-free supraspinatus in both groups. In other words, why the therapy rehabilitation with electromyographic feedback was not more efficient in increasing RMS value of supraspinatus of the pain-free shoulder?
Could the pain on dominant or non-dominant shoulder have different outcomes after therapy? Is there evidence on this matter? Would this impact the results and can it be considered a study limitation?
In line 254-256, the paragraph seems a justification for the study and it would fit better into the introduction.
Author Response
Dear Sir,
thank You so much for Your comments, which certainly will improve the quality of our paper.
Introduction
Q1: The introduction justified the relevance for studying shoulder rehabilitation therapies in wheel chair basketball players, but the rationale for why the therapy with electromyographic activity feedback would be more efficient is not provided. Additionally, there is not a hypothesis for the possible outcomes.
A1: we agree with this comment, so we integrated the introduction according to your requests.
Methods
Q1: The experimental design seems adequate, but the procedures are not well described and important details are missing. This lack of information precludes a deeper analysis.
There is no other information about the exercise protocol that the participants underwent besides the frequency of twice a week for 3 months. What was the session duration, exercise type, exercise volume, exercise intensity and the adherence of the therapy rehabilitation program throughout the experimental period?
A1: we integrated the description of the exercise protocol.
Q2: The use of mDurance system was poorly described, there is no information about the technical characteristics, configuration, practical application and handling of the equipment for the specific study. Moreover, the information regarding the signal data processing is also missing.
A2: we agree also with this comment, so we better described mDurance components and functioning.
Q3: It is not clear how the feedback provided by the surface EMG was used to implement the supraspinatus activity. In other words, what was the objective difference between the rehabilitation session carried out in the two groups? What was the variable visualized in real time (RMS) during the sessions? What was the parameter used to guide the therapy sessions? The same procedures were carried out in both shoulders or the procedures where different in the shoulder with and without pain?
A3: the difference consisted in the fact that just the Exercise Plus sEMG Biofeedback Group visualized in real time supraspinatus RMS as a colored column on the tablet screen during the sessions, so as that the exercises execution was specifically finalized to improve supraspinatus activity and to do it symmetrically for both shoulders. In this way, physiotherapist and patient observed and more rapidly resolved motor dyskinesias, normalizing the function of the shoulder affected by pain using the contralateral shoulder as a parameter.
Q4: The statistical analysis was adequate.
Results
The results are clearly presented.
A4: Thank You for these comments.
Discussion
Q1: The discussion is a “reprising” of the results. The study would benefit from a possible physiological mechanism to explain the therapeutic exercise effect on the rehabilitation of the outcomes. Moreover, it would be nice to find an explanation for the similar RMS values of the pain-free supraspinatus in both groups. In other words, why the therapy rehabilitation with electromyographic feedback was not more efficient in increasing RMS value of supraspinatus of the pain-free shoulder?
A1: This is an interesting question. As we stated in the text and as it is showed in the figures, the pain-free shoulders had a higher value of supraspinatus RMS at the baseline, since they did not present any type of evident dysfunction or alteration. Nonetheless, the exercises specifically dedicated to improve the activity of the supraspinatus in a symmetrical way in any case produced an increase in the RMS value of this muscle even on the pain-free shoulders, in this way effectively making them symmetrical with respect to the contralateral ones.
Q2: Could the pain on dominant or non-dominant shoulder have different outcomes after therapy? Is there evidence on this matter? Would this impact the results and can it be considered a study limitation?
A2: This is also a very interesting question. We exclude that the pain on dominant or non-dominant shoulder could affect the results, since the exercises were deliberately carried out in a symmetrical manner and above all they were aimed at professional athletes, i.e. called upon to train and use both shoulders in the same way. Moreover, the distribution of SP per side was homogeneous into the groups at the baseline, for this further reason we think that this aspect could not affect the results.
Q3: In line 254-256, the paragraph seems a justification for the study and it would fit better into the introduction.
A3: We agree, so we did the change that You suggested.
Best regards
Reviewer 3 Report
I congratulate the authors for this interesting work. Below, I make several comments in order to improve the manuscript.
The introduction is very limited. It is appreciated that the bibliography provided is current, but the introduction should be expanded to more adequately frame the object of study.
In Materials and Methods should be separated into different sections: Participants, Instruments, Procedure and Statistical Analysis.
They should provide more information about the participants, such as mean age, age range, years of experience in the sport, time injured, specific characteristics of the injuries, etc.
Because of the type of study, the procedure should be explained in much greater detail. The criteria used to assign the participants to each group should be indicated.
There is a need for a more in-depth discussion of the findings of this study, and to provide a more comparative analysis, highlighting the extent to which the current results support, extend, contrast, or challenge previous research. A clear explanation of how the current findings contribute to the literature and specific practical applications is lacking. What the results provide in terms of practical applications in sport needs to be further explored.
It is necessary that you check all references. Some follow the APA format and others follow the journal format. They should also enter the doi in the publications that require it.
Author Response
Dear Sir,
thank You so much for Your congratulations, which make us proud of our work, and for Your comments, which certainly will improve the quality of our paper.
Q1: The introduction is very limited. It is appreciated that the bibliography provided is current, but the introduction should be expanded to more adequately frame the object of study.
A1: We agree with this comment, so we integrated the introduction in order to better frame the object of the study.
Q1: In Materials and Methods should be separated into different sections: Participants, Instruments, Procedure and Statistical Analysis.
A1: We divided material and methods into different sections, as You required.
Q2: They should provide more information about the participants, such as mean age, age range, years of experience in the sport, time injured, specific characteristics of the injuries, etc.
A2: The information about age and age range are already described in table 1. Among the inclusion criteria, there was the following request: “sporting practice for at least 2 years”; we did not specifically deepen the years of sport experience, since this data does not affect the appearance of SP, which is frequent in all WB players, at all levels and regardless of the years of practice. With regard to the injuries, SP had to be present for at least a month but by definition it was an overuse injury, typical for this sport and in general for wheelchair users in the absence of trauma or other specific triggering events.
Q3: Because of the type of study, the procedure should be explained in much greater detail. The criteria used to assign the participants to each group should be indicated.
A3: We provided to better explain the procedures. Although it was not possible to perform a rigorous randomization of the subjects into the two groups due to the fact that the sample was of convenience, the characteristics of the participants were so peculiar that the groups were homogeneous for all evaluation parameters at baseline, as can be seen in table 1.
Q1:There is a need for a more in-depth discussion of the findings of this study, and to provide a more comparative analysis, highlighting the extent to which the current results support, extend, contrast, or challenge previous research. A clear explanation of how the current findings contribute to the literature and specific practical applications is lacking. What the results provide in terms of practical applications in sport needs to be further explored.
A1: we tried to expand the discussion section according to Your request.
Q1: It is necessary that you check all references. Some follow the APA format and others follow the journal format. They should also enter the doi in the publications that require it.
A1: we corrected the bibliography as You required.
Best regards
Round 2
Reviewer 2 Report
I would like to thank the authors for the effort to attend to my requests. The authors have done a good job addressing most of my questions and comments. Although the improvements of the current version of the manuscript, there remains some topics which I judged that still need to be more detailed.
I recognized the merits of the study and all my statements are meant to improve even more the study. I am mainly concerned with the methods description and interpretation. Therefore, I would like to suggest the inclusion of the following information:
The anatomical coordinates for positioning the bipolar sensor electrode used to register the supraspinatus sEMG as well as any proceedings related to the electrode handling (e.g. cleaning of the skin).
The range of weights and the characteristics of the rubber bands that were used in resistance exercise, the set of exercises that were executed at the therapeutic sessions (e.g. shoulder abduction, flexion, horizontal adduction)
In my opinion, the adaption of the volume and intensity of exercise within a pain-free range may be a subjective method since the final exercise load may vary significantly due to individual differences as the type and level of the lesion, pain tolerance, physical conditioning and previous experiences. Moreover, the lack of a quantitative parameter or an objective methodology to conduct the therapeutic session may difficult the protocol reproducibility. If the authors agree, I would like to suggest the inclusion of the above observations as a study limitation and to suggest the exercise individualization as a future direction to improve this innovative therapeutic approach.
P.S.: I understand that the pain-free shoulders had a higher value of supraspinatus RMS at the baseline, but the evidence that the electromyographic feedback was not more efficient on the pain-free shoulder still intrigues me. Maybe it could indicate that this therapeutic approach should be conducted unilaterally to be effective.
Author Response
Dear reviewer,
first of all we want to thank you also for your further comments. Your intent to improve our work is evident, therefore we really appreciate all your suggestions, therefore we want to try to comply with all your requests.
So, in the paper we described the electrodes positioning procedure and the exercises characteristics.
We agree with your suggestion to highlight explicitly the limitations due to the exercise protocol.
With regard to the pain-free shoulder, we believe that the results obtained, as you have correctly noted, are due to the fact that the shoulder affected by pain was already better trained than the contralateral shoulder, since in any case for the free-pain shoulders the progression towards better EMG values ​​was minor at the start since the available range was smaller. Anyway, we hope that this aspect can also be further investigated, perhaps by verifying only WB players without shoulder pain undergoing specific training for the shoulder.
Thank You again
Best regards